# Using Large Language Models for Hyperparameter Optimization

## Abstract

This paper explores the use of foundational large language models (LLMs) in hyperparameter optimization (HPO). Hyperparameters are critical in determining the effectiveness of machine learning models, yet their optimization often relies on manual approaches in limited-budget settings. By prompting LLMs with dataset and model descriptions, we develop a methodology where LLMs suggest hyperparameter configurations, which are iteratively refined based on model performance. Our empirical evaluations on standard benchmarks reveal that within constrained search budgets, LLMs can match or outperform traditional HPO methods like Bayesian optimization across different models on standard benchmarks. Furthermore, we propose to treat the code specifying our model as a hyperparameter, which the LLM outputs and affords greater flexibility than existing HPO approaches.

## 1 Introduction

Hyperparameters, distinct from the parameters directly learned throughout training, significantly influence the inductive bias of a machine learning model, thus determining its capacity to generalize effectively. The choice of hyperparameters helps dictate the complexity of the model, the strength of regularization, the optimization strategy, the loss function, and more. Due to this importance, many methods have been proposed to find good hyperparameter configurations automatically. For example, black-box optimization methods such as random search (Bergstra & Bengio, 2012) or Bayesian optimization (Mockus, 1998; Shahriari et al., 2015) have been deployed for hyperparameter optimization (HPO). Beyond black-box methods, other works use problem structure, e.g., are compute-budget-aware (Lam et al., 2016), multi-task (Swersky et al., 2013), multi-fidelity (Klein et al., 2017; Li et al., 2017), online (Jaderberg et al., 2017), or multi-objective (Daulton et al., 2022). However, these methods (a) still rely on practitioners to design a search space, which includes selecting which parameters can be optimized and specifying bounds on these parameters, and (b) typically struggle in the initial search phase ($< 2^d$ queries for $d$-dimensional hyperparameters). As such, tuning hyperparameters can be challenging for those with constrained budgets or without machine learning expertise, and methods that make HPO easier are desirable.

This paper investigates the ability of large language models (LLMs) to optimize hyperparameters. LLMs are trained on internet-scale data and have demonstrated emergent capabilities in new settings (Brown et al., 2020; OpenAI, 2023). We prompt LLMs with an initial set of instructions—describing the specific dataset, model, and hyperparameters—to recommend hyperparameters to evaluate. We train the model based on the proposed hyperparameters, record the final metric (e.g., validation loss), and ask the LLM to suggest the next set of hyperparameters. Figure 1 illustrates this iterative process. We found that LLMs were effective at proposing hyperparameters during initial search phases, making them valuable both for limited-budget scenarios and as a complement to traditional approaches when larger search budgets are available.

Our experiments demonstrated that LLMs can effectively optimize hyperparameters across a range of tasks, from toy problems to real-world applications. We first assessed LLMs' ability to optimize 2D toy objectives, where they received loss $\mathcal{L}(\boldsymbol{x})$ at specific points $\boldsymbol{x}$. Across a range of objectives, we found that LLMs effectively minimized the loss, exploiting performant regions while exploring untested areas. Next, to analyze whether LLMs could optimize realistic HPO settings, we evaluated our approach on standard HPO benchmarks

(Eggensperger et al., 2021), comparing it to common methods such as random search and Bayesian optimization. With small search budgets (e.g., 30 evaluations), LLMs improved upon traditional hyperparameter tuning. We also assessed the importance of using chain-of-thought reasoning (Wei et al., 2022b) and demonstrated that LLMs performed well over longer horizons of up to 100 evaluations. Additionally, LLMs were significantly faster than random search in tuning hyperparameters for both Vision Transformers and ResNets on the CIFAR-10 dataset, even when equipped with generic, architecture-agnostic prompts ("You are helping tune hyperparameters for a neural network.").

We further explored hyperparameter optimization using the unique flexibility that natural language enables. Rather than adhering to a fixed set of hyperparameter configurations, we prompted LLMs to produce training code (e.g., in PyTorch) to improve validation performance. This extension alleviates the need for human specification of the hyperparameters and their search spaces. The generated code can reduce the initial search for configuration spaces that are unlikely to succeed. With a limited search budget (5 evaluations), our results show that code generation performs better than random search and is comparable to LLM search with human-specified configurations. Finally, we conclude by discussing the potential of language models as general-purpose hyperparameter tuning assistants, limitations, and future work.

## 2 Method

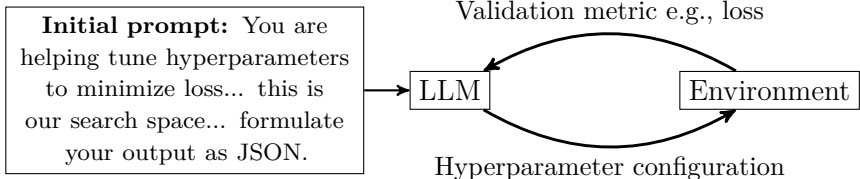

Figure 1: LLMs for hyperparameter optimization. We prompt an LLM with the problem description and the search space. The LLM then outputs a set of hyperparameters to evaluate. The environment, e.g., practitioner or automatic script, executes a training run with the hyperparameter setting, and then a validation metric is used to prompt the language model again.

### 2.1 Background

Hyperparameter optimization can be formulated as a bilevel optimization problem:

$$\boldsymbol{\lambda}^{\star} = \arg\min_{\boldsymbol{\lambda}} \mathcal{L}_V^* \left( \boldsymbol{\lambda} \right) = \arg\min_{\boldsymbol{\lambda}} \mathcal{L}_V \left( \boldsymbol{\lambda}, \mathbf{w}^{\star} \left( \boldsymbol{\lambda} \right) \right)$$
$$\text{s.t. } \mathbf{w}^{\star} \left( \boldsymbol{\lambda} \right) = \arg\min_{\mathbf{w}} \mathcal{L}_T \left( \boldsymbol{\lambda}, \mathbf{w} \right),$$

where $\mathcal{L}_T$ and $\mathcal{L}_V$ are training and validation objectives, and $\boldsymbol{\lambda}$ and $\mathbf{w}$ are hyperparameters and the model parameters, respectively. The objective aims to find the hyperparameters that minimize the validation loss when the training objective is trained to convergence.[1]

Hyperparameter optimization can often be performed sequentially, where a proposal $\boldsymbol{\lambda}_n$ depends on the sequence of prior values $\{\boldsymbol{\lambda}_1, \boldsymbol{\lambda}_2, \ldots, \boldsymbol{\lambda}_{n-1}\}$ and their validation losses. For example, Bayesian optimization (Mockus, 1998; Shahriari et al., 2015) builds a probabilistic model, such as a Gaussian process, to map hyperparameters $\boldsymbol{\lambda}$ to the validation loss $\mathcal{L}_V(\boldsymbol{\lambda}, \mathbf{w}^*(\boldsymbol{\lambda}))$. This approach iteratively selects the next hyperparameters to evaluate by optimizing an acquisition function that balances exploration and exploitation, thus converging to the optimal hyperparameters $\boldsymbol{\lambda}^*$ that minimize the validation loss. In a limited budget setting, practitioners often employ a trial-and-error "manual" search, choosing hyperparameters based on prior knowledge or experience (Yang & Shami, 2020). In this paper, we assess the ability of large language models in this role, hypothesizing they will be effective because they have been trained on internet-scale data and demonstrated emergent capabilities in new settings (Brown et al., 2020; Wei et al., 2022a).

---

[1]This assumes unique solutions for simplicity. See Vicol et al. (2022) for effects of non-uniqueness.

## 2.2 LLMs for Hyperparameter Optimization

We now describe our approach in more detail. When the hyperparameter search space is known, we prompt an LLM with a description of the problem and the search space. The LLM outputs a hyperparameter setting to evaluate. We then repeatedly alternate between two steps: (1) evaluating the current hyperparameter setting and (2) prompting the LLM with the validation metric (e.g., loss or accuracy) to receive the next set of hyperparameters. This continues until our search budget is exhausted. We illustrate this process in Figure 1 and list our prompts in Appendices B, D, and E. The prompts provide a high-level overview of the machine learning problem setup, the hyperparameters to be tuned, and, if given, their search spaces. This generality enables our approach to be applied to many domains.

```
### 1. Chat prompting
User: <<Machine learning problem description >>. Provide a config in JSON
      format. Config:
(Assistant: <<generated hyperparameter configuration >>
User: The validation loss was <<L>>. Provide the next config in the same
      format.) x N

### 2. Single message prompting
User: <<Machine learning problem description >>. Here is what we have tried
      so far:
<<Config 1>> <<Loss 1>>
<<Config 2>> <<Loss 2>>
...
<<Config N>> <<Loss N>>
Provide the next config in JSON format. Config:
```

Figure 2: Two ways to prompt the language model. Angular brackets vary with the problem or are dependent on what was generated in previous steps. Note that both approaches end with a user message so that the language model generates the next response.

More concretely, we consider two ways to prompt the model, following the popular "chat" interface (Touvron et al., 2023a; OpenAI, 2023) where users prompt the model with a dialogue consisting of "user" and "assistant" messages, depicted in Figure 2. The models we use generate tokens sampled according to an input temperature, terminating at a stop token. In the first approach, we prompt the model with the entire conversational history of messages, so the number of messages sent scales linearly with the number of steps. This happens because every new step contains two additional messages: the previous LLM response and the validation metrics from executing a training run from the hyperparameters in the response. The inference cost is linear (with a constant attention window) or quadratic in the number of tokens for a Transformer (Brown et al., 2020). For every new iteration, the number of tokens increases by the tokens representing the validation metrics and configuration. We denote this approach the "chat" prompt.

An alternative prompting strategy is to compress the search history into a single initial message that contains the problem description and lists the hyperparameter configuration history and corresponding validation metrics. This approach provides a more compact state representation, especially if we use chain-of-thought prompting (Wei et al., 2022b) to elicit reasoning, as in some experiments. The inference cost on compressed messages is cheaper than in the prior approach. Empirically, we observe that the two approaches achieve similar performance. We denote this approach as the "compressed" prompt and provide an implementation and example usage in our supplementary material.

When the search space is unknown, we can prompt an LLM to generate code for the model and optimizer, effectively providing a hyperparameter setting to evaluate. In this case, the environment, e.g., the practitioner, executes the generated code and provides the resulting output to the LLM. We use the prompt in Appendix D that asks the LLM to write code in PyTorch code that defines a model and optimizer. If the code fails to

produce a valid output, we can re-prompt with the error message. Code generation can also be viewed as hyperparameter tuning, where the hyperparameter $\boldsymbol{\lambda}$ is the input string representing the code. This extension alleviates the need for human specification of the hyperparameters to tune and their search spaces.

## 3 Related Work

**Hyperparameter Optimization (HPO).** Finding the optimal hyperparameter settings is crucial to achieving strong performance in machine learning (Bergstra & Bengio, 2012; Snoek et al., 2012). We refer readers to Feurer & Hutter (2019) and Bischl et al. (2023) for a general introduction to HPO. Initial research mainly explored model-free techniques such as grid and random search (Bergstra & Bengio, 2012). More advanced methods leverage multi-fidelity optimization – often by using that our optimization is an iterative process. For example, Hyperband (Li et al., 2017) and the Successive Halving (Jamieson & Talwalkar, 2016) introduced multi-armed bandit approaches to allow for the early stopping of less promising hyperparameter configurations. These methods are easy to parallelize and have shown promising results in practice, but they are dependent on random processes and do not fully leverage the HPO structure.

Bayesian Optimization (BO) (Mockus, 1998; Hutter et al., 2011; Snoek et al., 2012; Bergstra et al., 2013; Shahriari et al., 2015) builds a surrogate model from past function evaluations to choose promising future candidates. Bayesian optimization further models uncertainty, leveraged by optimizing an acquisition function instead of the loss directly. Since the initial successes of BO in optimizing hyperparameters, numerous tools have been developed to optimize the pipeline for improved efficiency and adaptability (Bakshy et al., 2018; Kandasamy et al., 2020; Balandat et al., 2020; Lindauer et al., 2022). However, Bayesian optimization is heavily influenced by the choice of surrogate model parameters (Lindauer et al., 2019), and faces scalability problems with an increased hyperparameter number being tuned or the number of past function evaluations. We show in our experiments that LLMs can exceed the performance of BO in the initial phase of tuning hyperparameters and the potential benefit of adopting a hybrid strategy of two algorithms.

Gradient or conditioning-based methods (Maclaurin et al., 2015; Franceschi et al., 2017; Lorraine & Duvenaud, 2018; MacKay et al., 2019; Bae & Grosse, 2020; Lorraine et al., 2020; Raghu et al., 2021; Bae et al., 2022; Mehta et al., 2024; Lorraine, 2024) are more scalable and efficient HPO methods. Nevertheless, they are often challenging to implement, require a differentiable objective, and must be deployed in the same location as the underlying model, making them less appealing general HPO solvers. A related method is OptFormer (Chen et al., 2022), which finetunes transformers on a large offline dataset to transfer learn a surrogate for gradient-based HPO.

**Decision making with LLMs.** Large language models (LLMs) have proven to be valuable in a variety of practical domains (Hariri, 2023; Liu et al., 2023), showing surprising emergent abilities, including in-context learning and chain-of-thought reasoning (Brown et al., 2020; Wei et al., 2022b; OpenAI, 2023). Although LLMs are known to occasionally give confident but incorrect answers (Ji et al., 2023), they are also shown to have reasoning capabilities, especially when explicitly guided to (Nye et al., 2021; Wei et al., 2022b; Kojima et al., 2022). Recent studies have used LLMs for optimization (Yang et al., 2024), such as finding the best downstream task prompt. They conducted experiments on linear regression, Traveling Salesman, and prompt optimization. In contrast, our work focuses on using LLMs to make decisions during hyperparameter optimization for machine learning tasks, including classical ML algorithms and training deep neural networks.

Others have integrated LLMs into the general AutoML pipeline (Chen et al., 2023; Hollmann et al., 2023; Lorraine et al., 2022; Zhang et al., 2023; Zheng et al., 2023). For example, Zheng et al. (2023) demonstrate that LLMs can find competitive architectures on neural architecture search benchmarks (Su et al., 2021).

## 4 Results

We evaluated our approach through a series of experiments. First, we assessed performance using the HPOBench benchmark (Eggensperger et al., 2021), where we tune hyperparameters across eight datasets for four different models: neural networks, SVMs, random forests, and logistic regression. We then used LLMs to tune hyperparameters for more complex architectures by optimizing Vision Transformers and

ResNets on CIFAR-10. We also demonstrate how our method generalizes to code generation tasks. To build intuition about how LLMs navigate the optimization landscape, we analyzed their behavior on CIFAR-10 and low-dimensional problems in Appendix E, where we can directly visualize the optimization trajectories.

## 4.1 Hyperparameter Optimization on HPOBench

We report results on the HPOBench (Eggensperger et al., 2021) benchmark, which defines search spaces for various models on publicly available OpenML AutoML benchmarks. Eggensperger et al. (2021) showed that their benchmarks yield different hyperparameter-loss landscapes and are effective for benchmarking algorithm performance. We used the first 8 datasets provided and tuned hyperparameters for the algorithms implemented in Sklearn: logistic regression, support vector machines (SVMs), random forests, and neural networks (Pedregosa et al., 2011), resulting in 32 different tasks. As defined by the benchmark, we tune 2 hyperparameters for SVMs, 2 for logistic regression, 4 for random forests, and 5 for neural networks. Search space details can be found in Appendix B, as well as figures of results on each task. We compare the following HPO approaches:

**LLMs.** Our initial experiments use OpenAI's time-stamped versions of GPT-3.5 and GPT-4 models in: *gpt-3.5-turbo-0613* and *gpt-4-0613* with temperature 0. GPT-4 Turbo was released after our initial results, so we conducted experiments with the model *gpt-4-1106-preview* and found better performance at less than half the price of GPT-4.[2]

**Random.** Random search often outperforms grid and manual search, thriving when the hyperparameter space has low effective dimensionality (Bergstra & Bengio, 2012). The design spaces of a given benchmark dictate where we should sample – e.g., log-uniformly for regularization hyperparameters in $\mathbb{R}^+$ or from a discrete set such as the number of neural network layers.

**Bayesian Optimization (BO).** We evaluate two Bayesian HPO methods in the SMAC3 library (Lindauer et al., 2022) that use random forest and Gaussian processes as surrogate models. We use the default recommended settings, including an initial design of up to 25% of the optimization budget.

Table 1: We summarize the performance of various HPO Algorithms on 8 datasets and 4 search spaces tuning hyperparameters for logistic regression, SVMs, random forests, and neural networks. For all tasks, we use 10 function evaluations. As summary metrics, we report how often each method beats random search (GPT-4 Turbo beats random search 81.25% of the time). We also compute the change in validation error for each optimizer versus random search and report the median and mean change across 32 tasks. The mean rank is computed between the 5 HPO approaches and random search, i.e., each method is assigned a rank between 1 and 6 on a task. The mean rank for random across the 32 tasks is 4.00.

| Model | Versus Random ($\uparrow$) | Median Change ($\uparrow$) | Mean Change ($\uparrow$) | Mean Rank ($\downarrow$) |
|---|---|---|---|---|
| GPT-4 Turbo | 81.25% | 13.70% | 19.83% | 2.42 |
| GPT-4 | 68.75% | 4.58% | 8.54% | 3.48 |
| GPT-3.5 Turbo | 43.75% | -0.82% | -13.58% | 3.84 |
| Bayes Opt$_{RF}$ | 56.25% | 2.11% | 5.86% | 3.45 |
| Bayes Opt$_{GP}$ | 50.00% | -0.01% | -8.28% | 3.80 |

In Table 1, we report aggregated results comparing two LLMs, random search, and the two BO algorithms. We compare the difference in validation error versus random search (0.2 to 0.1 is a 50% improvement). Every algorithm has a search budget of 10 evaluations. For random search, we randomly sample 500 configurations for each model and dataset combination, evaluate the corresponding losses, and then use a bootstrapped sample to estimate the loss on a given budget. As a summary metric, we report the mean and median improvement in the 32 tasks of each algorithm versus random search. We show the performance on each dataset and model in Figure 7 and Figure 8. GPT-4 Turbo beats random search most frequently, achieving the highest mean and median improvement. When we compare the mean rank for the five search algorithms across the 32 tasks, GPT-4 Turbo is again consistently better.

---

[2]OpenAI says "preview" means *gpt-4-1106-preview* is unsuited for production traffic. We did not encounter problems.

Table 2: Chain-of-thought reasoning prompting may offer modest improvement versus prompting a model to immediately output the next set of hyperparameters. We often saw reasoning that describes the effects of adjusting various hyperparameters.

| Model | Beats Random ($\uparrow$) | Median Change ($\uparrow$) | Mean Change ($\uparrow$) |
|---|---|---|---|
| GPT-4 Turbo CoT | 81.25% | 13.70% | 19.83% |
| GPT-4 Turbo | 81.25% | 15.58% | 21.23% |
| GPT-4 CoT | 68.75% | 4.58% | 8.54% |
| GPT-4 | 65.62% | 3.43% | 11.55% |
| GPT-3.5 Turbo CoT | 43.75% | -0.82% | -13.58% |
| GPT-3.5 Turbo | 40.62% | -3.19% | -117.12% |

Table 3: Evaluating performance on a longer trajectory of 30 function calls. GPT-4-Turbo still maintains good performance, suggesting that it is capable of selecting problem-specific hyperparameter settings. The mean rank between all 5 methods (each row + random) is 3.

| Model | Versus Random ($\uparrow$) | Median Change ($\uparrow$) | Mean Change ($\uparrow$) | Rank ($\downarrow$) |
|---|---|---|---|---|
| GPT-4 Turbo CoT | 78.12% | 12.10% | 18.27 % | 2.72 |
| GPT-4 Turbo | 90.62% | 18.94% | 22.34 % | 2.22 |
| Bayes Opt$_{RF}$ | 71.88% | 5.91% | 14.63 % | 3.06 |
| Bayes Opt$_{GP}$ | 84.38% | 11.80% | 21.07 % | 2.75 |

**How important is chain-of-thought reasoning?**  In Table 2, we evaluate the effects of chain-of-thought (Wei et al., 2022b) on performance. To elicit reasoning, our intermediate messages to the language model are:

```
loss = {loss:.4e} Write two lines as follows:
Analysis: Up to a few sentences describing what worked so far and what to
    choose next
Config: (JSON config)
```

The loss is expressed in scientific notation to four decimal places. See Appendix B for complete details on the prompts. We see that including reasoning has a large positive impact on GPT-3.5 and a marginal impact on GPT-4. Reasoning chains can be informative for a practitioner at the cost of additional tokens. We show a snippet of GPT-4 reasoning when tuning hyperparameters for a neural network:

- (loss 0.193) **GPT-4**: *The loss remained the same despite changes in the hyperparameters. This suggests that the model might not be sensitive to these parameters or has reached a local minimum. We should try a more drastic change in the learning rate and depth.*

- (loss 0.215) **GPT-4**: *The loss increased, indicating that the last configuration was not beneficial. The increase in learning rate and the decrease in depth might have caused the model to overshoot the minimum. Let's try reducing the learning rate and increasing the depth.*

There are potential concerns for the faithfulness of explanations (Lanham et al., 2023), though we observe consistent reasoning that trades off between exploration and selecting a strong final value. We also see logic that understands interactions and effects of different hyperparameters, presumably garnered from the LLM training on textbooks and other resources on HPO. We provide additional full trajectories in Appendix G.

**Longer trajectories**  We evaluated our approach on longer time horizons, with up to 100 proposed configurations, and show the results in Table 3 and Table 5. Here, we used the "compressed" prompt described in Section 2: we prompted the LLM with a single message at each step, showing the entire history of hyperparameter configurations and losses. A longer search trajectory has a reduced dependency on

initialization and requires a better search algorithm for good performance. GPT-4 Turbo still achieved a better average rank with hyperparameter optimization trajectories up to length 60 and is comparable in trajectories of length 100.

We observe the reasoning generated by the language model is a useful summary of experimental results and the relative importance of different hyperparameters. For example, when tuning a random forest, the LLM outputs the following reasoning:

- **GPT-4-Turbo**: *The lowest loss so far is with Config 9, which suggests that a combination of a slightly deeper tree (max_depth: 15), a moderate number of features (max_features: 0.5), and a higher number of samples at the leaf nodes (min_samples_leaf: 4) and for splitting (min_samples_split: 8) is beneficial. The loss has not improved significantly with changes to the max_depth beyond the default, so we should explore other parameters more. We can try increasing the min_samples_leaf further since increasing it from 1 to 4 improved the performance, and we can also experiment with a different max_features value to see if it affects the loss.*

We additionally ran a preliminary experiment to assess if LLMs can be useful to initialize Bayesian optimization—we find that for a search trajectory of length 30 total steps, using the GPT-4 Turbo proposed configurations for the first ten steps improves or matches performance on 21 of the 32 tasks (65.6%) for Bayesian optimization with a random forest surrogate function. This suggests LLMs can be synergistic with existing approaches.

## 4.2 CIFAR-10

To evaluate performance in a more difficult deep learning scenario, we tested our method on tuning Vision Transformers (ViTs) (Dosovitskiy et al., 2020) and ResNet-9 (He et al., 2016) on CIFAR-10. This setup poses a challenge for hyperparameter tuning since ViTs typically operate in the regime of large-scale pretraining. We tune five hyperparameters: optimizer choice, learning rate, batch size, weight decay, and label smoothing. To make the task more difficult, we limit training to just 20 epochs and provide minimal context to the LLM, prompting with "You are helping tune hyperparameters for a neural network" without any details about the architecture or dataset. We use the compressed prompting format without chain-of-thought.

Despite these constraints, the LLM successfully optimized validation loss, outperforming random search as shown in Figure 3. Again, we focus on the initial search phase because LLM-based approaches can leverage learned information about hyperparameters to make better proposals. We evaluated two approaches for hyperparameter specification from the LLM: (1) using the same structured search space as random search and (2) allowing free-form valid hyperparameter suggestions, such as requiring batch size to be a positive integer but not restricting its range. The second approach provides more flexibility while maintaining validity constraints. Full experimental details are provided in Appendix C.

## 4.3 Ablation Studies on Prompt Information and Robustness

We conducted experiments on varying the amount of information presented in the prompt while still tuning a Vision Transformer with the hyperparameter search space above. These are plotted in Figure 4.

**What is the effect of specifying additional details about the model and dataset?** We investigated how providing varying levels of detail in the initial prompt affects hyperparameter tuning performance for Vision Transformers. In addition to the base prompt, we considered three conditions:

1. Adding dataset information ("...tune hyperparameters for CIFAR-10")

2. Adding both dataset and correct architecture ("...tune a Vision Transformer on CIFAR-10")

3. Adding dataset and incorrect architecture ("...tune a convolutional neural network on CIFAR-10")

While all conditions reach similar performance by iteration 10 (loss $\approx 0.65$), the amount of information provided can improve hyperparameter selection in the first two steps.

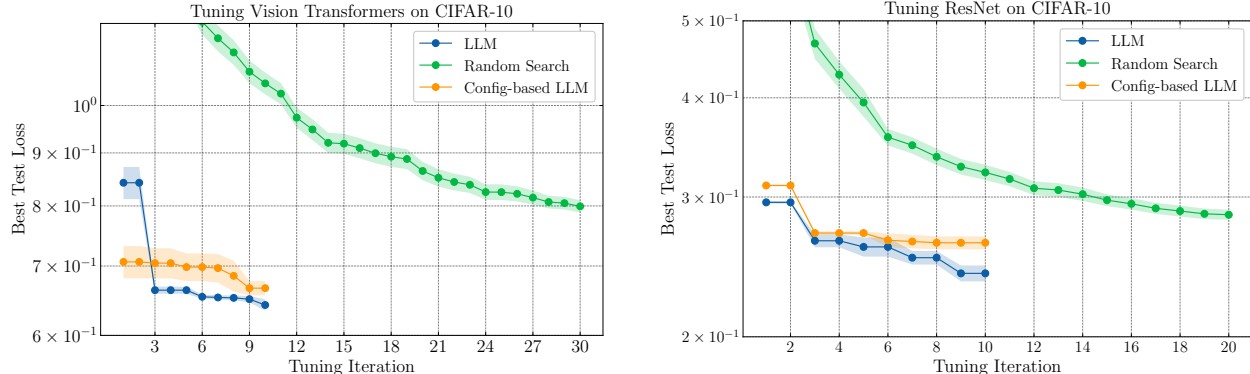

Figure 3: Performance comparison of hyperparameter optimization methods on CIFAR-10. Left: Tuning Vision Transformers shows LLM-based approaches achieve lower validation loss compared to random search after 30 iterations. The config-based LLM approach, which uses explicit hyperparameter ranges, performs similarly to the unconstrained LLM. Right: Similar results for ResNet architecture. The best validation loss is tracked across iterations to reflect real-world tuning scenarios.

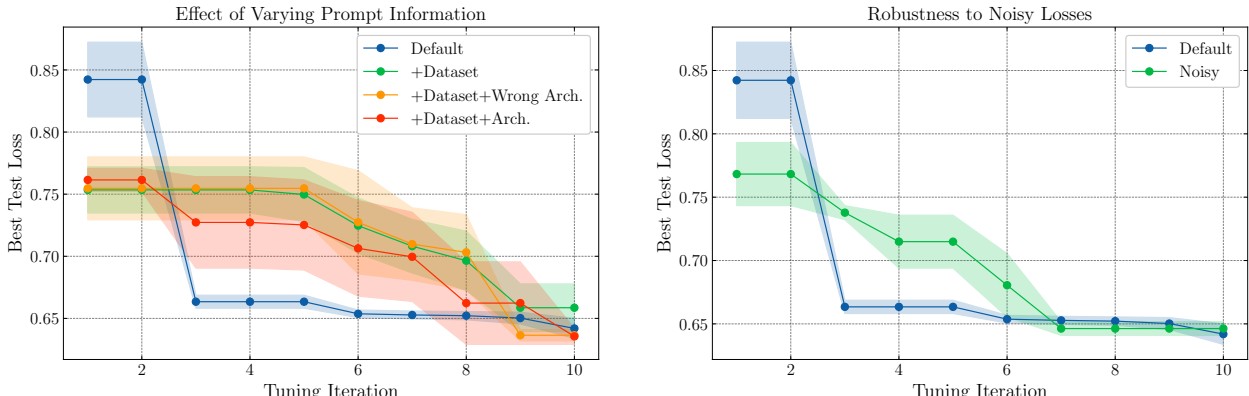

Figure 4: Effect of prompt information and measurement noise on hyperparameter optimization for Vision Transformers. Left: Best validation loss across tuning iterations with varying levels of initial prompt detail, from basic instructions to including dataset and architecture information. The amount of information provided can improve hyperparameter selection in the first two steps but all conditions reach similar performance at iteration 10. Right: Comparison of optimization performance with clean versus noisy ($\pm 10\%$) loss measurements. The similar performance suggests robustness to measurement noise.

**How robust is the LLM to noisy measurements?** We found that LLM performance was similar even when there was noise in the optimization process. Here, we multiplied the loss and accuracy by a random value sampled uniformly from $(0.9, 1.1)$ at each step before passing it to the LLM. This suggests that LLMs can be robust to noisy measurements in the inner optimization loop, which is important since the process of training neural networks can have stochasticity in validation metrics.

**Visualizing the Search Trajectories** We randomly sampled two hyperparameter search trajectories and visualized the evolution of the batch size and the learning rate in Figure 5. In the trajectories we examined, we found a reasonable hyperparameter search strategy. There is evidence of conditional hyperparameter selection, where the LLM selects a learning rate for SGD higher than all its choices for Adam, which follows conventional wisdom. SGD is used only once, which may be appropriate in this small-budget problem. Bigger batch sizes are not tried after a batch size of 256, which resulted in a higher loss.

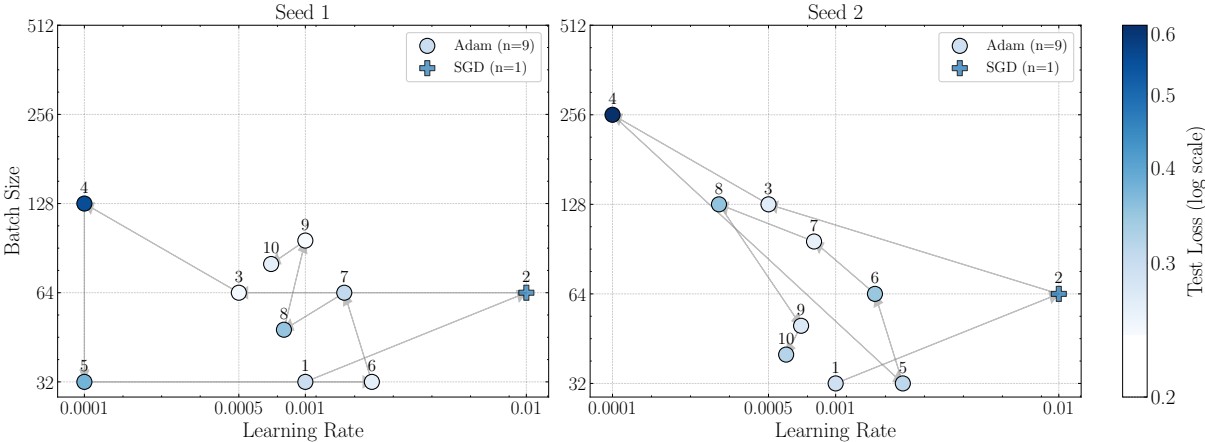

Figure 5: Visualization of hyperparameter optimization trajectories from the GPT-4 tuning ResNet. GPT-4 selected SGD once (denoted with a plus sign) and Adam with the remaining proposals in both training runs. The learning rate is adjusted accordingly to the optimizer and regions with high loss were not revisited.

## 4.4 Code generation

Code generation offers a flexible paradigm to specify the configuration space for hyperparameter tuning. In this setup, we treat the model and optimizer source code as hyperparameters tuned by the LLM. We use code generation to avoid needing to specify a configuration space to search over. The code generation approach is evaluated against random search and LLMs with a fixed, configuration-based search space. Our evaluation included both GPT-4 Turbo (OpenAI, 2023) and Llama-3.3 (Dubey et al., 2024) models operating within a fixed configuration search space. The baseline methods are provided with a search space, including the network's depth and width, batch size, learning rate, and weight decay. To minimize the risk of data leakage issues where the LLM was trained on performant hyperparameter settings for our dataset, we used a Kaggle NYC Taxi dataset (Taxi & Commission, 2023a;b) for our experiments that was released after the knowledge cutoff of the LLM. We provide our prompts and additional details in Appendix D.

In the first tuning iteration, we asked the LLM (GPT-4 Turbo) to output model and optimizer source code, which implicitly requires specifying the hyperparameters. This code is generated as functions that take in hyperparameters as arguments. Hyperparameter tuning is performed in later iterations by asking the LLM to generate function calls with specific arguments. The model is run with the given settings, and the LLM receives the validation loss and the average training losses during each epoch as feedback.

Table 4 reports the minimum test loss achieved using a fixed search budget of 5 evaluations, simulating the initial search phase results during hyperparameter tuning. We randomly sample 200 configurations and use a bootstrapped sample to estimate the standard error for a random search. Figure 6 (bottom) reports the minimum test loss found at each step and illustrates that code generation obtains better initial settings than competing approaches.

Table 4: Minimum test loss after 5 tuning iterations. We report the mean and standard error for code generation and the config-based LLM across 5 runs.

| Method | Min Test Loss ($\downarrow$) |
|---|---|
| Random search | $3.757 \times 10^{-3} \pm 1.172 \times 10^{-3}$ |
| Code generation | $2.754 \times 10^{-4} \pm 9.241 \times 10^{-5}$ |
| Config-based LLM (GPT-4 Turbo) | $1.218 \times 10^{-4} \pm 2.959 \times 10^{-5}$ |
| Config-based LLM (Llama-3.3 70B) | $1.127 \times 10^{-4} \pm 3.789 \times 10^{-5}$ |
| Bayes Opt$_{RF}$ | $2.130 \times 10^{-4} \pm 1.026 \times 10^{-4}$ |

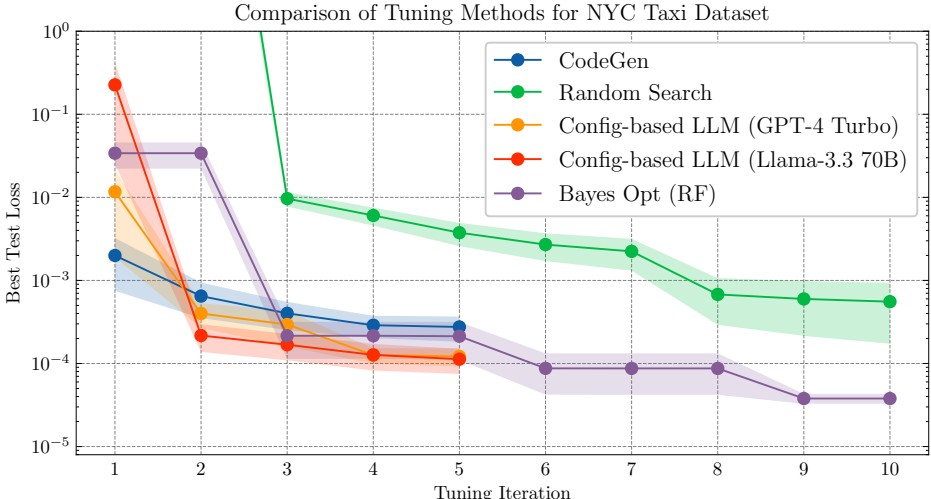

Figure 6: Test loss trajectory comparison with a standard error over 5 runs on NYC Taxi. The initial proposal and early iterations from direct code generation and configuration-based LLM search compare favorably to random search and Bayes Optimization with a random forest surrogate.

## 5   Discussion

We discuss limitations and explore broader implications and future work.

**Possibility of Dataset Contamination.**   While some evaluation benchmarks may exist in LLM training data, our prompting process only provides function outputs without revealing benchmark details, reducing the possibility of direct memorization. Previous research suggests that such contamination might not critically influence overall performance (Radford et al., 2021). Building more private data sets unseen by LLM during training would be valuable for future studies or even synthetic datasets like our quadratic experiments.

**Challenges with Reproducibility.**   The exact inference procedures for LLMs like GPT-4 (OpenAI, 2023) are not publicly disclosed, making replicating the results potentially challenging. While setting the sampling temperature to 0 should theoretically yield deterministic outputs, this is not always observed in practice. For future studies, running more experiments with open source LLMs (Taori et al., 2023; Touvron et al., 2023a;b) to establish a benchmark with reproducible results would be beneficial.

**Cost.**   It can potentially be costly to conduct full-scale HPO experiments with GPT-4. For example, executing 32 tasks with trajectory 10 using chain-of-thought reasoning on HPOBench costs approximately $8 USD as of March 2024. This cost can increase further when using more iterations but can be reduced using the "compressed" prompt. For instance, ten rounds of hyperparameter tuning with the "compressed" prompt and GPT-4, as in the ablation experiments, only costs five cents. These efficiency benefits of using an LLM for HPO could be worthwhile if the cost of the underlying experiment is expensive. Additionally, in Figure 6, we show that our method demonstrates consistent performance with open-source models like Llama-3.3 (Dubey et al., 2024), offering the potential for even greater cost reduction through self-hosting.

### 5.1   Future Directions

Our experiments demonstrate LLMs' quantitative and qualitative effectiveness for HPO but also pose a new conundrum for how this ability emerges. Wei et al. (2022a) suggest one hypothesis for emergent abilities: "more parameters and more training enable better memorization, which could be helpful for tasks requiring world knowledge." This may partially explain the jump in performance we observe from GPT-3.5 to GPT-4, but the underlying mechanisms for this emergence are not yet well understood.

**LLMs as Research Assistants**   Initial experiments demonstrated promising applications beyond simple hyperparameter suggestion. While traditional optimization methods focus solely on parameter selection, as shown in Figure 10, LLMs provided useful feedback for error messages through natural language interaction.

However, models can face characteristic challenges of current language models, including hallucinations (Bowman, 2023). For instance, in Figure 11, we observed the LLM making incorrect assumptions about model architecture and suggesting potentially inappropriate regularization strategies.

Our findings suggest several promising research directions. First, exploring how to ground LLM suggestions in verified technical knowledge could improve reliability. Second, developing more robust dialogue systems for experimental debugging could enhance the assistant paradigm. The future may lie in more interactive and intuitive optimization processes that maintain human oversight while making use of the general knowledge of language models.

### 5.2 Broader Impact Statement

The approach presented in this paper facilitates machine learning research and applications by potentially making it easier to find well-performing hyperparameters. Using LLMs as hyperparameter tuners may help lower the cost of model training (e.g., time, compute, and environmental impact), making machine learning experiments easier for those from other disciplines. Overall, the benefits and risks are likely similar to those of other automatic machine learning (AutoML) research. Code generation with LLMs should be done with care; Zelikman et al. (2023) presented an overview of potential risks and trade-offs.

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

## A  Appendix

Our appendix includes details for the HPOBench experiments in Section B and code generation in Section D. We also include 2-dimensional landscape experiments in Section E, dialogues of using LLMs as Tuning Assistants in Section F, and trajectories of LLM responses in Section G.

## B  HPOBench

### B.1  Prompts

Our initial prompt is the concatenation of a generic message and the Sklearn documentation for specific hyperparameters. We used the same initial message for all models. Our prompt is the concatenation of beginning, middle (model-dependent), and end.

**beginning** You are helping tune hyperparameters for a {model}. Training is done with Sklearn. This is our hyperparameter search space:

These search space descriptions are copied verbatim from Sklearn and the benchmark configuration.

**middle**

**svm** C, regularization parameter. The strength of the regularization is inversely proportional to C. Must be strictly positive. The penalty is a squared l2 penalty. Type: UniformFloat, Range: [0.0009765625, 1024.0], Default: 1.0, on log-scale
gamma, Kernel coefficient for rbf, Type: UniformFloat, Range: [0.0009765625, 1024.0], Default: 0.1, on log-scale

**logistic regression** alpha, constant that multiplies the regularization term. The higher the value, the stronger the regularization. Type: UniformFloat, Range: [1e-05, 1.0], Default: 0.001, on log-scale
eta0, The initial learning rate for the adaptive schedule. Type: UniformFloat, Range: [1e-05, 1.0], Default: 0.01, on log-scale

**random forest** max_depth, the maximum depth of the tree. Type: UniformInteger, Range: [1, 50], Default: 10, on log-scale
max_features, the number of features to consider when looking for the best split. Type: UniformFloat, Range: [0.0, 1.0], Default: 0.5

min_samples_leaf, the minimum number of samples required to be at a leaf node. Type: UniformInteger, Range: [1, 20], Default: 1

min_samples_split, the minimum number of samples required to split an internal node. Type: UniformInteger, Range: [2, 128], Default: 32, on log-scale

**neural net** alpha, l2 regularization, Type: UniformFloat, Range: [1e-08, 1.0], Default: 0.001, on log-scale

batch_size, Type: UniformInteger, Range: [4, 256], Default: 32, on log-scale

depth, Type: UniformInteger, Range: [1, 3], Default: 3

learning_rate_init, Type: UniformFloat, Range: [1e-05, 1.0], Default: 0.001, on log-scale

width, Type: UniformInteger, Range: [16, 1024], Default: 64, on log-scale

**end** We have a budget to try 10 configurations in total. You will get the validation error rate (1 - accuracy) before you need to specify the next configuration. The goal is to find the configuration that minimizes the error rate with the given budget, so you should explore different parts of the search space if the loss is not changing. Provide a config in JSON format. Do not put new lines or any extra characters in the response. Example config: {"C": x, "gamma": y} Config:

## B.2 Transition Messages

After receiving the initial config, we evaluate the hyperparameters and prompt the language model with the following:

```
loss = {loss:.4e}. Specify the next config, do not add anything else in your
    response. Config:
```

Here "{loss:.4e}" represents the loss in scientific notation with 4 decimal places.

After that, we use one of the following prompts, where the validation loss is updated based on the result of the training run.

```
### Chain of Thought (CoT)
loss = {loss:.4e} Write two lines as follows:
Analysis: Up to a few sentences describing what worked so far and
what to choose next
Config: (JSON config)

### Normal (No CoT)
loss = {loss:.4e}. Specify the next config.
```

Finally, on the last attempt before our budget is exhausted, we preface the message with "This is the last try." These prompts were not tuned for the task — we did not adjust them beyond adding "do not add anything else" to ensure that the language model output is easily parsable. Our reported results are a single seed with a temperature of 0 for the LLM.

## B.3 Other Experiment Details

We use the default arguments for `HyperparameterOptimizationFacade` and `BlackBoxFacade` in the SMAC library (Lindauer et al., 2022), which correspond to using Random Forest and Gaussian Process surrogate models with BO.

### B.4   Results

In Figure 8, we show the validation accuracy achieved on each task after 10 updates. In Table 5, we show results over trajectories of 60 iterations and 100 iterations. GPT-4 Turbo consistently beats random search and is comparable to Bayesian Optimization approaches.

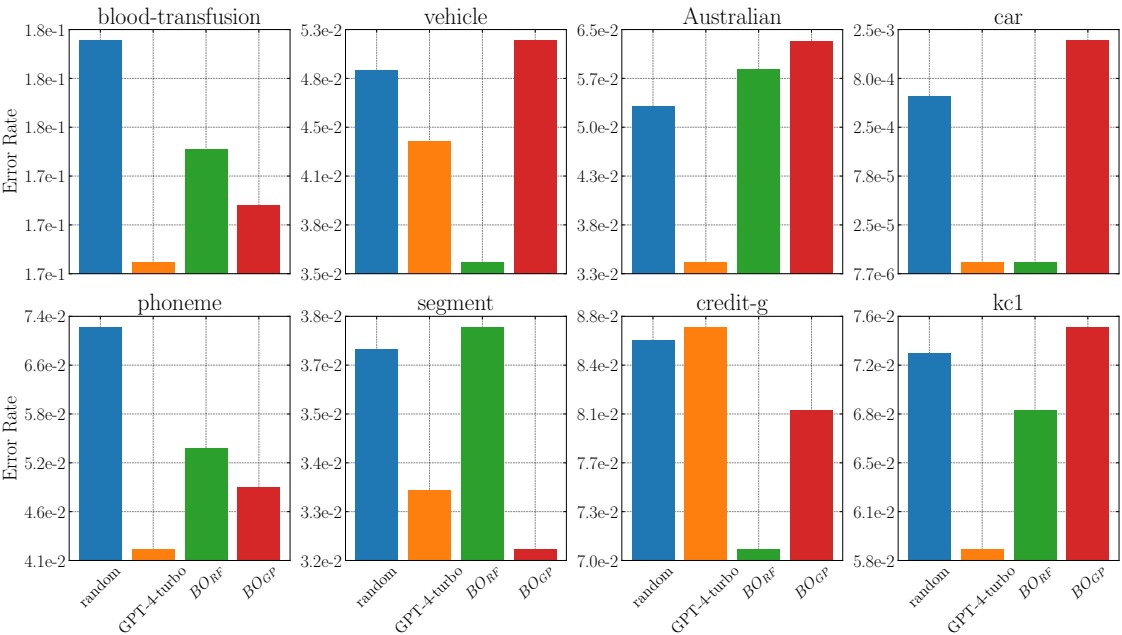

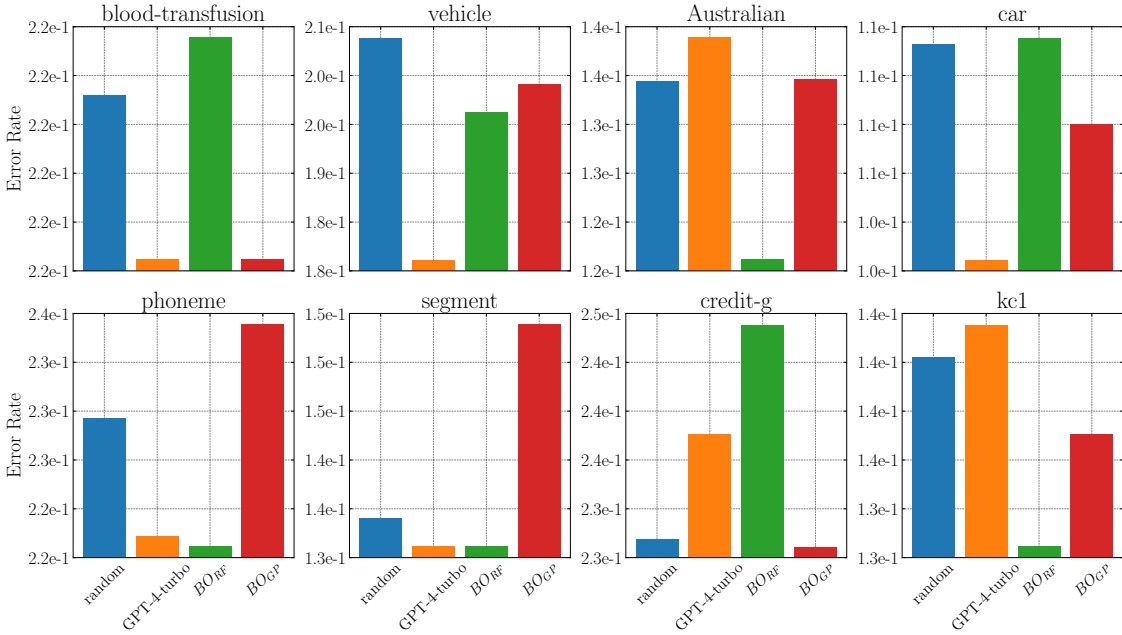

Figure 7: Minimum validation error achieved after 10 function evaluations from different hyperparameter optimizers on datasets from HPOBench (Eggensperger et al., 2021). The benchmark defines a configuration search space on neural networks for learning rates, $\ell_2$ regularization, width, depth, and batch size. A large language model (GPT-4 Turbo) performs well compared to random search and Bayesian optimization ($BO$) with Gaussian process ($GP$) and random forest ($RF$) surrogate models. Aggregate results across four models are in Table 1.

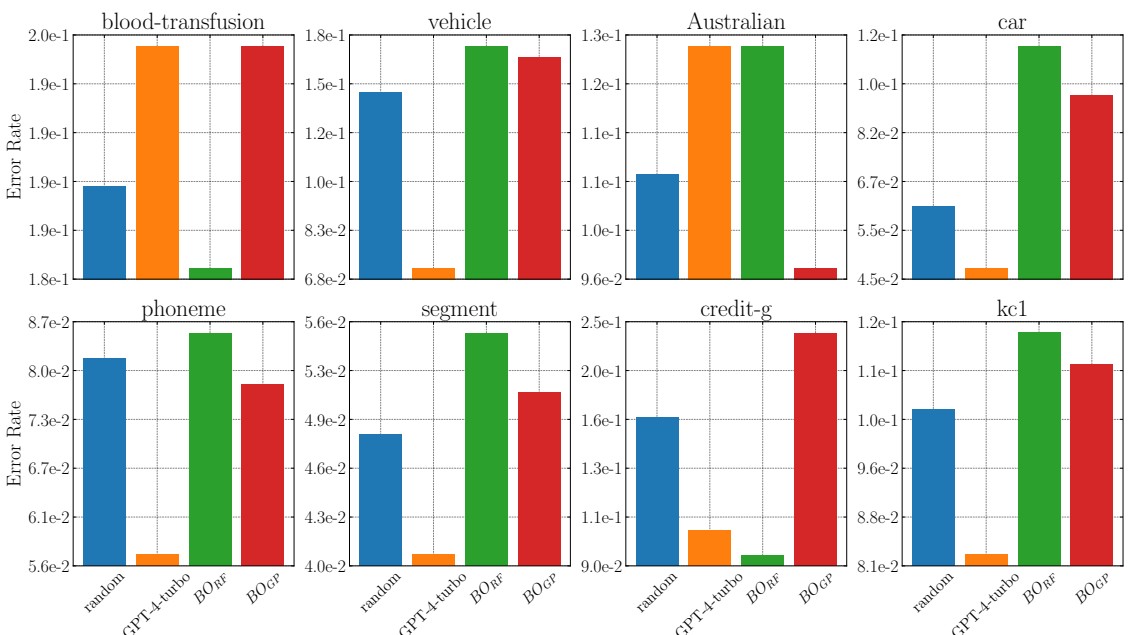

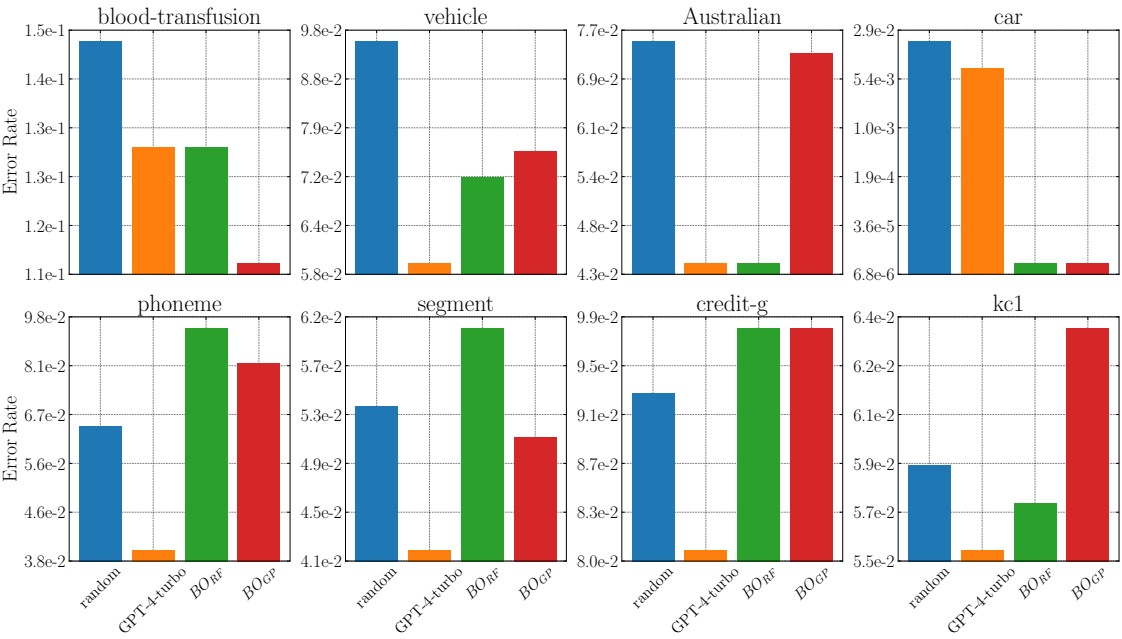

Figure 8: Minimum validation error achieved after 10 function evaluations from different hyperparameter optimizers on datasets from HPOBench (Eggensperger et al., 2021). We show performance on each task tuning Random Forests and SVMs. Aggregate results across four models are in Table 1.

Table 5: Evaluating minimum validation error achieved for trajectories of length 60 and 100. We consider tuning hyperparameters for SVMs, random forests, and logistic regression trained on eight datasets for a total of 24 tasks. Neural networks were too costly to train for trajectories of this length, so we used the other three models. GPT-4-Turbo consistently outperforms random search and is comparable to Bayesian optimization up to trajectories of length 100. The mean rank between all 4 methods (each row + random) is 2.5.

| 60 iterations | | | | |
| --- | --- | --- | --- | --- |
| Model | Beats Random ($\uparrow$) | Median ($\uparrow$) | Mean ($\uparrow$) | Rank ($\downarrow$) |
| GPT-4 Turbo | 87.50% | 16.31% | 22.76 % | 1.94 |
| $BO_{RF}$ | 87.50% | 19.61% | 22.57 % | 2.21 |
| $BO_{GP}$ | 91.67% | 14.36% | 24.78 % | 2.19 |
| 100 iterations | | | | |
| Model | Beats Random ($\uparrow$) | Median ($\uparrow$) | Mean ($\uparrow$) | Rank ($\downarrow$) |
| GPT-4 Turbo | 87.50% | 11.71% | 21.88 % | 2.08 |
| $BO_{RF}$ | 91.67% | 14.15% | 19.69 % | 2.33 |
| $BO_{GP}$ | 91.67% | 12.75% | 23.89 % | 1.88 |

### B.5 Additional Results

We consider the effect of appending a system prompt "You are a machine learning expert" in Table 6. We use this prompt throughout the rest of the experiments, following the idea of conditioning on good performance (Zhou et al., 2022).

Table 6: Evaluating effects of system prompt. We report the mean and median change as in the main paper.

| Model | Beats Random ($\uparrow$) | Median ($\uparrow$) | Mean ($\uparrow$) |
| --- | --- | --- | --- |
| Expert | 90.62% | 7.75% | 23.23 % |
| Non-expert | 81.25% | 13.79% | 22.36 % |

## C CIFAR-10 Experiment Details

The Vision Transformer (ViT) model we use has an input image size of 32x32 pixels, divided into patches of 4x4. It has a hidden dimension of 512, a depth of 6 layers, and 8 attention heads. The model has MLP dimensiosn of 512, and both dropout and embedding dropout rates are set to 0.1.

We ran each experiment with three random seeds at temperature 0 for the LLM. We only compute the test accuracy on the 10 000 points once, at the end of 20 training epochs. We use the compressed prompting approach with the following prompt:

```
You are helping tune hyperparameters for a neural network. This is our
    hyperparameter search space:
{
    "optimizer": must be ["adam", "sgd"]
    "learning_rate":  between 1e-4 and 1e-1
    "train_batch_size": 32, 64, 128, 256, 512
    "weight_decay": between 1e-5 and 1e-1
    "label_smoothing": between 0 and 0.2
}
You will get the validation error rate and loss before you need to specify
    the next configuration. The goal is to find the configuration that
    minimizes the error rate with the given budget, so you should explore
    different parts of the search space if the loss is not changing.
```

```
        Provide a config in JSON format. Do not put new lines or any extra
        characters in the response, only provide the config. Example config:
{
    "optimizer": a
    "learning_rate": b
    "training batch size": c
    "weight_decay": d
    "label_smoothing": e
}
```

Random search samples from the same search space, with learning rate and weight decay sampled log-uniformly. We evaluate 100 samples and use bootstrapping to estimate the mean and standard error for plots. The unconstrained LLM prompt updates the first part of the prompt to:

```
"optimizer": must be ["adam", "sgd"]
"learning_rate": positive float
"train_batch_size": positive integer
"weight_decay": nonnegative float
"label_smoothing": nonnegative float
```

## D  Code Generation

This is the initial prompt we use for code generation:

```
I'm going to use your abilities to generate a model and tune its
    hyperparameters such that it performs well on a Kaggle challenge.

{dataset.problem_description}

You have {dataset.in_features} input features.
The names of the features in the dataset are: {dataset.X_columns}
The target variable is: {dataset.y_columns[0]}

You may assume that the dataset already preprocessed and available as a
    DataLoader object.

I want you to write PyTorch code that creates a model and optimizer.
Write everything in one function that is called 'make_model_and_optimizer'.
Later I will ask you to call this function so make sure it does not
    reference any global variables.
Also make sure that you include all the hyperparameters that you anticipate
    needing to tune later as arguments into this function.
Favour using primitive or built-in types and avoid using Callable or Module
    types as arguments to this function.
Be sure to include a short docstring and type annotations.

Finally, write a short sentence explaining your reasoning.

Format your output as follows:

reasoning: <your reasoning here>

code:
'''python
```

```
<your code here>
'''
```

The `make_model_and_optimizer` function is then parsed and validated. This is done by first checking that it is valid Python code, and then by checking its function signature to make sure that the function arguments and the outputs are as specified. If these constraints are not met, the error is fed back into the LLM, and it regenerates the code.

The agent now tunes hyperparameters it has specified for this function by making calls to it. This is done by taking advantage of the function calling feature[3] in the OpenAI API to get our LLM to directly output a call into this function.

Here is the tuning prompt:

```
Now let's tune the hyperparameters of your model.

Give me instances of a both the model and an optimizer by making
a call to 'make_model_and_optimizer'.
Use hyperparameters that you think will perform well on the validation set.
I will then train the model and give you feedback on how well it performs.
You have {search_budget} iterations to tune your model.
Your respone should just be one function call to
'make_model_and_optimizer'.
```

After training the model with the proposed settings, we provide feedback to the model using the following prompt:

```
Here is the output of your code:

training loss over each epoch:
{', '.join([f'{l:.3f}' for l in feedback.train_losses])}
validation loss: {feedback.val_loss:.3f}

Based on this, make a new call to 'make_model_and_optimizer'
with hyperparameters that you think will perform better
on the validation set.
```

The baseline methods are given the following configuration space to optimize:

```
alpha, l2 regularization, Type: UniformFloat, Range: [1e-08, 1.0], Default:
      0.001, on log-scale
batch_size, Type: UniformInteger, Range: [4, 256], Default: 32, on log-scale
depth, Type: UniformInteger, Range: [1, 3], Default: 3
learning_rate_init, Type: UniformFloat, Range: [1e-05, 1.0], Default: 0.001,
      on log-scale
width, Type: UniformInteger, Range: [16, 1024], Default: 64, on log-scale
```

## E   2-dimensional Landscape Experiment

To evaluate whether LLMs can reason about optimization choices, we study a set of 2-dimensional toy test functions commonly used in optimization: Rosenbrock (Rosenbrock, 1960), Branin (Dixon, 1978),

---

[3]https://platform.openai.com/docs/guides/function-calling.

Himmelblau (Himmelblau et al., 2018), and Ackley (Ackley, 1987). For each test function $f$, we consider optimizing $f(\boldsymbol{x})$ and also $f(\boldsymbol{x} - \boldsymbol{c})$, where $\boldsymbol{c}$ is a fixed constant, with $\boldsymbol{c}_i \sim \mathcal{U}(0, 1)$, to mitigate the LLM's potential to memorize the original $f$. Also, we benchmark performance on a well-conditioned and ill-conditioned quadratic. We provide the experimental details and the explored prompts in Appendix E.

We search for optimal solutions using LLMs as an optimizer with GPT-4. While these problems can be considered black-box optimization, we can also pose them as HPO to the LLM, using the same prompts in Figure 1 for consistency. We show the performance of a single fixed seed and temperature 0 in Figure 9. Performance across 3 seeds outperforms random search in most tasks, as in Table 8. Moreover, inspecting the trajectories and reasoning chains shown in Appendix G, we qualitatively observe different behaviors that may characterize a strong search algorithm: initial exploration around search space boundary and center; a line-search type algorithm to probe function behavior; and reasoning that trades off between exploration and selecting a performant final value. However, performance can be inconsistent between random seeds, as shown in Table 7.

We use prompts with minimal details to gain intuition on this HPO approach and evaluate our LLM approach on problems with less complexity. Inspecting the trajectories, we see room for more specific prompts. For example, the LLMs re-explored previously evaluated points. We obtain the same or better performance in 8 of the 10 tasks by adding a sentence stating that the functions are deterministic, shown in Table 9. We speculate that HPO performance can generally be improved with more problem-setting details in the prompt.

### E.1 Prompts

Prompt 0 is a practical description of the problem as black-box optimization.

```
You are optimizing a function with two inputs. x1 must be in range
    {search_space['x1']\. x2 must be in range {search_space['x2']}. I want
    you to predict values that minimize the loss of the function; I will
    tell you the value of the function before you try again. Do not put new
    lines or extra characters in your response. Format your output with
    json as follows: {{"x": [x1, x2]}}
```

Prompt 1 adds one sentence to include the compute budget.

```
You are optimizing a function with two inputs. x1 must be in range
    {search_space['x1']}. x2 must be in range {search_space['x2']}. I want
    you to predict values that minimize the loss of the function; I will
    tell you the value of the function before you try again. Do not put new
    lines or extra characters in your response. We have a total of {budget}
    evaluations. Format your output with json as follows: {{"x": [x1,
    x2]}}\n
```

Prompt 2 is similar to the second but frames the problem as hyperparameter optimization.

```
You are helping tune hyperparameters to minimize loss. x1 must be in range
    {search_space['x1']}. x2 must be in range {search_space['x2']}. I want
    you to predict values that minimize the loss of the function; I will
    tell you the value of the function before you try again. Do not put new
    lines or extra characters in your response. We have a total of {budget}
    evaluations. Format your output with json as follows: {{"x": [x1,
    x2]}}\n
```

We also study Prompt 3, adding a sentence: "The training process is deterministic and yields a nonnegative loss."

| Toy Function | Prompt 0 | Prompt 1 | Prompt 2 | Temp 0 |
|---|---|---|---|---|
| ack | 0.0, 0.0, 0.0 | 0.0, 0.0, 0.0 | 0.0, 0.0, 0.0 | 0.0, 0.0, 0.0 |
| shifted_ack | 1.81, 2.52, 2.83 | 2.19, 2.19, 2.19 | 4.37, 2.83, 1.77 | 1.77, 1.86, 4.48 |
| bran | 10.82, 2.88, 3.09 | 1.03, 9.87, 1.94 | 9.87, 9.87, 0.5 | 1.94, 3.91, 0.5 |
| shifted_bran | 15.54, 15.8, 18.68 | 7.79, 1.04, 3.26 | 2.39, 9.9, 50.04 | 3.26, 5.58, 9.9 |
| rosen | 1.0, 1.0, 1.0 | 1.0, 0.0, 1.0 | 1.0, 0.0, 1.0 | 0.0, 1.0, 0.0 |
| shifted_rosen | 7.03, 7.03, 5.3 | 0.78, 167.43, 215.4 | 2.99, 215.4, 0.78 | 0.53, 163.86, 15.37 |
| himmel | 0.0, 8.12, 8.12 | 2.0, 8.12, 106.0 | 8.12, 8.12, 26.0 | 8.12, 8.12, 8.12 |
| shifted_himmel | 4.57, 4.57, 4.57 | 47.07, 14.46, 47.07 | 47.07, 152.1, 1.58 | 16.99, 47.07, 47.07 |
| quad2d | 0.44, 28.15, 0.44 | 11.24, 28.15, 12.75 | 0.72, 0.32, 0.32 | 1.45, 0.37, 28.15 |
| quad2d_illcond | 52.35, 51.45, 51.14 | 51.45, 51.45, 51.45 | 18.45, 0.65, 51.21 | 51.45, 51.14, 0.65 |

Table 7: Minimum loss achieved in 10 iterations on 3 different seeds at temperature 0.1. We also run the experiments for Prompt 2 with temperature 0. Note that there is randomness in the results even at temperature 0.

| Toy Function | Prompt 0 | Prompt 1 | Prompt 2 | Temp 0 | Random |
|---|---|---|---|---|---|
| ack | 0.0 | 0.0 | 0.0 | 0.0 | $5.28 \pm 1.73$ |
| shifted_ack | 2.39 | 2.19 | 2.99 | 2.7 | $5.26 \pm 1.74$ |
| bran | 5.6 | 4.28 | 6.75 | 2.12 | $5.83 \pm 5.57$ |
| shifted_bran | 16.68 | 4.03 | 20.78 | 6.25 | $7.13 \pm 6.00$ |
| rosen | 1.0 | 0.67 | 0.67 | 0.33 | $481.34 \pm 998.30$ |
| shifted_rosen | 6.46 | 127.87 | 73.06 | 59.92 | $618.82 \pm 1691.99$ |
| himmel | 5.42 | 38.71 | 14.08 | 8.12 | $20.39 \pm 19.93$ |
| shifted_himmel | 4.57 | 36.2 | 66.92 | 37.04 | $21.47 \pm 20.00$ |
| quad2d | 9.68 | 17.38 | 0.46 | 9.99 | $4.72 \pm 5.76$ |
| quad2d_illcond | 51.65 | 51.45 | 23.44 | 34.41 | $15.14 \pm 17.61$ |

Table 8: Average minimum loss achieved in 10 iterations on 3 different seeds at temperature 0.1. We also run the experiments for Prompt 2 with temperature 0. Note that there is randomness in the results even at temperature 0. We report the mean and standard deviation for random across 1000 trials

| Toy Function | Prompt 2 | Prompt 3 |
|---|---|---|
| ack | 0.0 | 0.0 |
| shifted_ack | 2.7 | 2.19 |
| bran | 2.12 | 3.57 |
| shifted_bran | 6.25 | 1.23 |
| rosen | 0.33 | 0.33 |
| shifted_rosen | 59.92 | 129.31 |
| himmel | 8.12 | 6.05 |
| shifted_himmel | 37.04 | 36.21 |
| quad2d | 9.99 | 0.87 |
| quad2d_illcond | 34.41 | 27.08 |

Table 9: Average minimum loss achieved in 10 iterations on 3 different seeds at temperature 0. Note that there is randomness in the results even at temperature 0. Prompt 3 adds a sentence "The training process is deterministic and yields a nonnegative loss." to Prompt 2. This achieves the same or better performance on 8 of the 10 tasks.

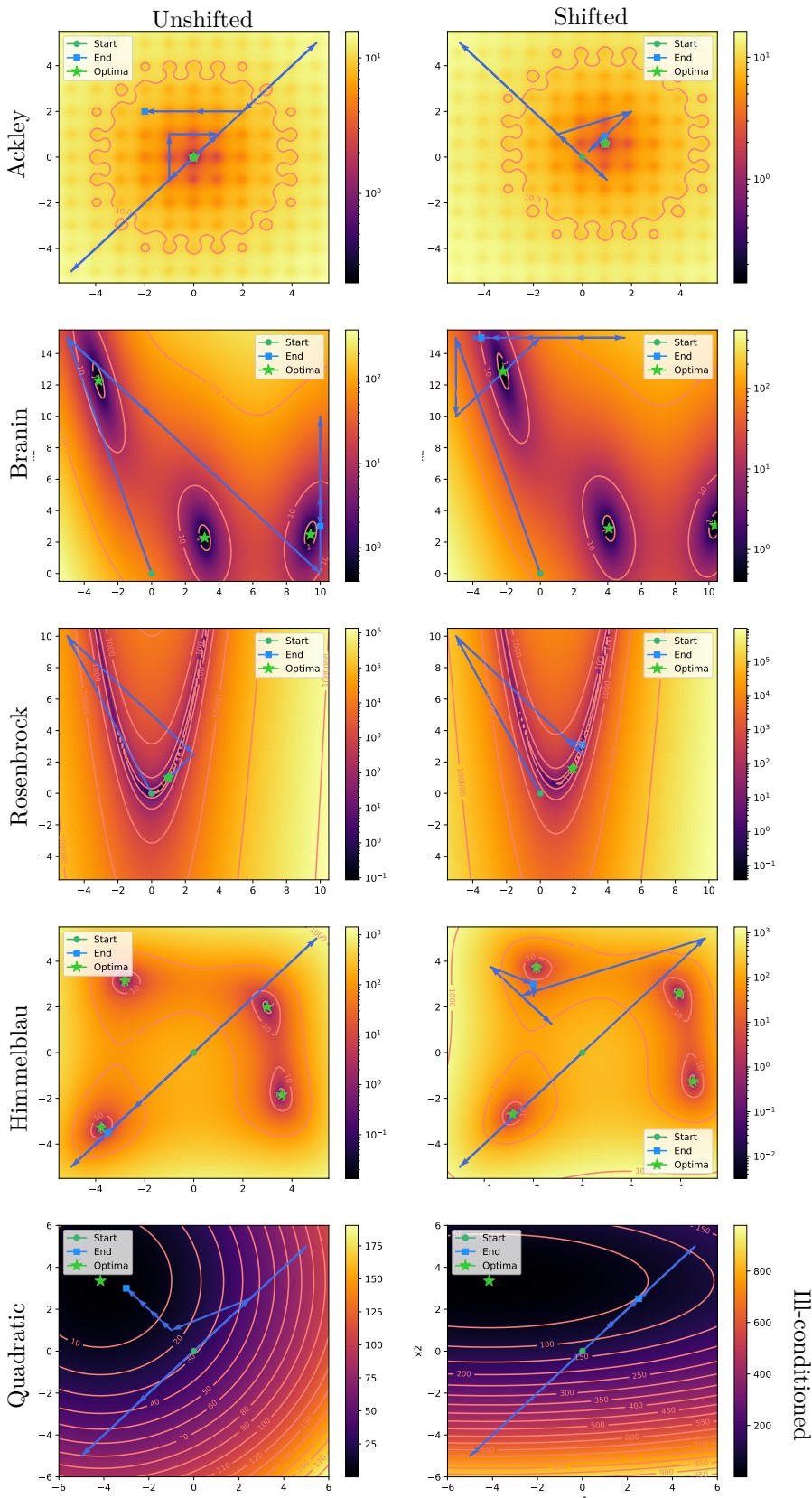

Figure 9: The LLM optimizer optimized the function in most situations with few function evaluations, even when we apply random argument shifts to mitigate memorization. Darker colors represent regions of lower (better) loss, with the optimum denoted by a star.

## F    LLMs as Tuning Assistants

Large language models offer exciting potential for hyperparameter tuning as they can offer interactive help to debug and improve models, using natural language. We highlight some potential applications and limitations. We show the temperature 0 responses from different language models.

In Figure 10, LLMs provide useful feedback to someone encountering an error message, which is infeasible with traditional approaches. A user can also seek additional clarification after implementing suggestions. In Figure 11, we show a potential shortcoming of this approach. The LLMs generated content assumes the user is using a neural network and also suggests using regularization (which may be unhelpful if the model is underfitting); an expert would potentially respond by first requesting clarification from the user (e.g. what model they are training, and specifics on validation and training performance).

**Question:** I'm using Adam with a learning rate of 0.1 to train a neural network and get a NaN error. What should I do?

**GPT-3.5-Turbo:** If you are encountering a NaN (Not a Number) error while training a neural network with Adam optimizer and a learning rate of 0.1, it typically indicates that the gradients have exploded or become too large. Here are a few steps you can take to address this issue:

1. Reduce the learning rate: A high learning rate can cause the gradients to become unstable and result in NaN errors. Try reducing the learning rate to a smaller value, such as 0.01 or 0.001, and see if the issue persists.

2. Check for numerical instability: NaN errors can also occur due to numerical instability in the network. Ensure that your input data is properly normalized and that there are no extreme values or outliers that could cause instability.

3. Implement gradient clipping: Gradient clipping is a technique that limits the magnitude of the gradients during training. By setting a maximum threshold for the gradient values, you can prevent them from becoming too large. This can help mitigate the NaN error. Most deep learning frameworks provide built-in functions for gradient clipping.

4. Check for bugs in your code: Review your code implementation to ensure there are no bugs or mistakes that could lead to NaN errors. Double-check the loss function, network architecture, and any custom layers or operations you have implemented.

5. Use regularization techniques: Regularization methods like L1 or L2 regularization, dropout, or batch normalization can help stabilize the training process and prevent NaN errors. Consider applying appropriate regularization techniques to your network.

6. Verify your data: NaN errors can also occur due to problematic data. Check your dataset for missing values, corrupted samples, or any other data-related issues that could lead to NaN errors.

7. Monitor the network's behavior: During training, keep an eye on the loss values and gradients. If you notice sudden spikes or extremely large values, it can indicate a problem. You can also try visualizing the gradients to identify any abnormalities.

By following these steps, you should be able to diagnose and address the NaN error in your neural network training.

**GPT-4:** Getting a NaN error during training usually indicates that your model is diverging, and the weights are becoming too large to represent. This is often due to a too high learning rate. Here are some steps you can take to resolve this issue:

1. Reduce the learning rate: A learning rate of 0.1 is quite high for the Adam optimizer. Try reducing it to 0.001 or 0.0001.

2. Gradient Clipping: If your model is still diverging after reducing the learning rate, you might want to try gradient clipping. This is a technique to prevent exploding gradients by capping them at a certain value.

3. Check your data: Make sure there are no NaN or infinite values in your input data. Also, check if your data is properly normalized. Large input values can also cause the model to diverge.

4. Check your model architecture: Make sure your model architecture is appropriate for your task. If your model is too complex, it might overfit and diverge.

5. Regularization: If your model is overfitting, you might want to add some form of regularization, like dropout or weight decay.

6. Check your loss function: Make sure your loss function is appropriate for your task and correctly implemented. Some loss functions can produce NaN values if not used correctly.

Remember, debugging a neural network often involves a lot of trial and error. Don't be discouraged if the first thing you try doesn't work. Keep experimenting and you'll eventually find a solution.

Figure 10: Response to question about NaN.

**Question:** I've trained a model but have low accuracies; what hyperparameters should I consider tuning?

**GPT-4:** 1. Learning Rate: This is one of the most important hyperparameters. If it's too high, the model might overshoot the optimal solution. If it's too low, the model might need too many iterations to converge to the best values. So, finding a good value is crucial.

2. Number of Epochs: This is the number of times the entire training set is shown to the network while training. Increase it if the model is underfitting.

3. Batch Size: It is the number of training examples used in one iteration. For a large dataset, it might be beneficial to use a larger batch size to speed up the training process, but it also might be less accurate.

4. Number of Layers: If your model is underfitting, you can try to make your model more complex by increasing the number of layers.

5. Number of Neurons: The number of neurons in a layer can also affect the model performance. You can try increasing this number for underfitting models.

6. Activation Function: The choice of activation function can have a big impact on the performance of the model. ReLU, sigmoid, and tanh are some commonly used activation functions.

7. Dropout Rate: Dropout is a regularization technique that prevents overfitting. If your model is overfitting, you can try increasing the dropout rate.

8. Weight Initialization: The way the initial weights are set in your neural network can have a big impact on the performance of the model.

9. Optimizer: Different optimizers can have a big impact on the performance of your model. Some common optimizers are SGD, Adam, RMSProp, etc.

10. Momentum (generally used with SGD optimizer): It helps in accelerating gradients vectors in the right directions, thus leading to faster converging.

Remember, tuning hyperparameters is a trial and error process. It's about understanding what each one does and how they can potentially impact your model training and accuracy.

Figure 11: Response to question about low accuracies. The LLM-generated answer assumes the user is using a neural network and also suggests using dropout, which generally does not help for underfitting models.

## G   Trajectories

### G.1   2D experiments

We provide code for loading the trajectories in `load_toy_trajectories` in our supplementary material.

## G.2   HPOBench

Here, we show some of the initial chain-of-thought reasoning for tuning neural networks on HPOBench generated by GPT-4.

```
Task 1
{"alpha": 0.001, "batch_size": 32, "depth": 3, "learning_rate_init": 0.001,
     "width": 64}

{"alpha": 0.01, "batch_size": 16, "depth": 2, "learning_rate_init": 0.01,
     "width": 256}

Analysis: So far, the initial configuration performed well with a loss of
     0.19283. The second configuration resulted in a slightly higher loss of
     0.19731. It seems that increasing the learning rate and reducing the
     batch size did not improve the performance.

Config: {"alpha": 0.001, "batch_size": 32, "depth": 3, "learning_rate_init":
     0.001, "width": 64}

Analysis: The previous configurations resulted in decreasing performance,
     with the latest configuration having a loss of 0.24664. It seems that
     the current hyperparameters are not optimal for minimizing the error
     rate.

Config: {"alpha": 0.1, "batch_size": 64, "depth": 1, "learning_rate_init":
     0.1, "width": 1024}Analysis: The latest configuration showed some
     improvement with a loss of 0.21525, compared to the previous
     configuration. It seems that reducing the depth and increasing the
     width of the network had a positive impact.

Config: {"alpha": 0.001, "batch_size": 16, "depth": 2, "learning_rate_init":
     0.001, "width": 512}Analysis: The latest configuration resulted in
     further improvement with a loss of 0.17937, indicating that reducing
     the depth and increasing the width of the network continues to be
     effective. It seems that a smaller batch size also contributed to the
     improved performance.

Config: {"alpha": 0.01, "batch_size": 8, "depth": 2, "learning_rate_init":
     0.01, "width": 512}Analysis: The latest configuration resulted in a
     loss of 0.19731, which is similar to the previous best configuration.
     It seems that further exploration is needed to find a better
     combination of hyperparameters.

Config: {"alpha": 0.001, "batch_size": 16, "depth": 3, "learning_rate_init":
     0.001, "width": 256}Analysis: The latest configuration resulted in a
     loss of 0.17937, which is the same as the previous best configuration.
     It seems that we have reached a plateau in terms of performance
     improvement.

Config: {"alpha": 0.001, "batch_size": 8, "depth": 2, "learning_rate_init":
     0.001, "width": 512}Analysis: The latest configuration resulted in a
     loss of 0.21525, which is slightly worse than the previous best
     configuration. It seems that reducing the depth and increasing the
     width of the network may not always lead to better performance.
```

