# OpenReview forum: "Using Large Language Models for Hyperparameter Optimization"
_TMLR — Rejected by TMLR_

### Review · Reviewer_mvVE · 2024-11-16

**Summary Of Contributions:**

(1) The authors claim to develop a novel method, in which they prompt LLMs with dataset and model descriptions which are iteratively refined based on model performance for the purpose of hyperparameter optimization. (2) The authors claim as a contribution the revelation that LLMs can match or outperform traditional HPO methods like Bayesian optimization across different models on standard benchmarks. (3) The authors propose to treat the code specifying their model as a hyperparameter.

**Audience:**

Yes

**Broader Impact Concerns:**

None.

**Claims And Evidence:**

No

**Requested Changes:**

The following three weaknesses are critical to securing my recommendation for acceptance --

* The baselines in Table 1 are weak, undercutting (2). Looking at this table, a reader might get the idea that no HPO method existed in the literature which could reliably outperform random chance, which is misleading and contrary to existing literature (https://arxiv.org/abs/2109.06716). To better enable comparisons between your method and prior work, please add at least three strong baseline methods, such as ASHA, BOHB, Hyperband, or recent works http://arxiv.org/abs/2402.03921, http://arxiv.org/abs/2305.15930. The HPOBench paper itself provides many options.
* HPOBench and the datasets it uses are now quite old. As you note in discussion, data contamination is a major challenge when evaluating LLMs on static benchmarks (https://iclr.cc/virtual/2024/poster/17911), undercutting (2). There is no reason to leave these experiments to future work. Please add to Table 1 at least two datasets past your LLM's knowledge cutoff date to see whether performance degrades. Adding stronger baseline methods to Table 4 (e.g., standard HPO methods over your random sample configurations) would also satisfy this note.
* As noted in your discussion section, closed source LLMs are near-complete black boxes, and make poor research baselines (https://hackingsemantics.xyz/2023/closed-baselines/), undercutting (1). Please demonstrate that your method works when using open-source models as well -- for example, the recent OpenCoder (https://github.com/OpenCoder-llm/OpenCoder-llm). This is not a challenging set of experiments to run, there is no reason to leave it to future work.

**Strengths And Weaknesses:**

The authors have supplied many interesting and helpful ablations. In particular, the ablations in Sec. 4.3 are revealing, as they indicate that the LLM barely even using the metadata provided to it during its search, suggesting strongly either that the solutions have been partially memorized by the LLM, or that the optimization problem is so easy that 'educated guesses' based on generic ML knowledge are sufficient to solve it. The LLM is also "robust" to noisy measurements, again suggesting it is probably making limited use of the information provided to it.

The idea of prompting an LLM to generate code for the model and optimizer when the search space is unknown is inventive.

For weaknesses, please see the requested changes.

---

> ### Author Response · Authors · 2024-12-16
>
> We thank the reviewer for the detailed feedback.
>
> > The baselines in Table 1 are weak, undercutting (2)
>
> The results in Table 1 showcase the strengths of our algorithm. As we emphasized in our abstract and introduction, our primary claim focuses on the ability of LLMs to propose good hyperparameters in "small" or "constrained" budget scenarios. These scenarios often rely heavily on human intuition to guide initial configurations.
>
> From our understanding of the [SMAC documentation](https://automl.github.io/SMAC3/main/3_getting_started.html), the initial sequences proposed by BO rely on either Sobol or random sequences, both of which we consider in Table 1. Over longer horizons, as we see in Table 3 and Table 5, both versions of BO consistently outperform random search. This aligns with existing literature and demonstrates that our approach complements, rather than replaces, BO. We can revise the manuscript to clarify this context to address your concern over the reader.
>
> With greater compute budgets, BO will perform better. However, the two approaches can be synergistic. Our experiment to assess if LLMs can be useful to initialize Bayesian optimization showed that performance was matched or improved on "21 of the 32 tasks (65.6%) for Bayesian optimization."
>
> > Contamination, open-source models and more baselines
>
> As mentioned in our paper, “our prompting process only provides function outputs without revealing benchmark details, reducing the possibility of direct memorization.”
>
> In response to your comment, we ran additional experiments on the Kaggle NYC taxi dataset, which contains taxi cab data from January to August 2023. We used the Llama 3.3-70b model, an open source model with performance comparable to GPT-4 Turbo. Additionally, we added a Bayesian optimization baseline to this table. The results show that our method performs significantly better under constrained search conditions.

---

### Review · Reviewer_1ZQj · 2024-12-02

**Summary Of Contributions:**

This paper explores the innovative use of large language models (LLMs) for hyperparameter tuning in machine learning models. Conventional methods, such as grid or random search, often become computationally prohibitive as the dimensionality of the search space increases. Hierarchical search and Bayesian optimization (and its variants) are widely used alternatives, but Bayesian optimization faces limitations with the number of past function evaluations and challenges in kernel design.

The authors conduct extensive experiments using the HPOBench framework, evaluating four models across eight datasets, resulting in a total of 32 tasks. They further extend the study to complex architectures, including Vision Transformers and ResNets on CIFAR-10, to demonstrate the approach's applicability. The paper examines the effect of chain-of-thought reasoning on hyperparameter tuning performance and performs ablation studies to evaluate the impact of prompt information and robustness to noise.

**Audience:**

Yes

**Broader Impact Concerns:**

Discussed in Section 5.2. I do not have further comments on this.

**Claims And Evidence:**

Yes

**Requested Changes:**

Overall, the paper is very well written, and I do not have any presentation-related suggestions.

The content is also quite clear. Using LLMs for hyperparameter tuning has been tested on a range of tasks and compared to random search and Bayesian optimization-based methods.

If the authors could comment on:

The limitations of this method, including when it does not work.
The internal mechanism of LLMs that enables them to solve the task of hyperparameter tuning.
I think these additions could strengthen the message and reception of the paper.

Minor comment:

Table 1 appears to be missing a caption.

**Strengths And Weaknesses:**

The main concern with a paper of this nature is the depth of contribution, which I believe is ultimately for the area chair and journal editors to decide. While the idea of using LLMs for hyperparameter tuning is interesting, it feels somewhat basic and closely resembles other applications of LLMs. Conceptually, this falls into the broader approach of taking a problem $P$ and evaluating whether LLMs can solve it.

The paper could be significantly strengthened by providing a deeper understanding of the limitations of the proposed method. For instance: What determines the difficulty of a hyperparameter tuning problem?
Under what conditions can we expect this method to succeed or fail?
What is the internal mechanism by which the LLM solves this problem? Does the LLM essentially fit a function from hyperparameter configuration to validation loss and make decisions based on that?
Without addressing such fundamental questions, the contribution risks being limited to simply applying LLMs to a new problem. Of course, this is just my perspective, and I acknowledge that these may be very challenging questions to answer.

On the positive side, the breadth of experiments conducted by the authors is a significant strength and lends credibility to the proposed approach. Additionally, the paper is clearly written and easy to follow.

---

> ### Author Response · Authors · 2024-12-16
>
> Thank you for your detailed and constructive feedback.
>
> > Deeper understanding of underlying mechanisms.
>
> We agree that understanding the internal mechanisms by which LLMs succeed at hyperparameter tuning is important. Based on our experiments in the main paper and the 2D toy experiments in Appendix E, we have some observations that can provide insights.
>
> For hyperparameter tuning, two important factors are the choice of initial hyperparameters and the search mechanism—how the optimizer selects the next set of hyperparameters based on previous results. Figure 5 shows that LLMs tend to initialize with configurations likely to perform well (e.g., choosing commonly used defaults such as Adam with a learning rate of 1e-3). Subsequently, they refine these configurations, probing regions of low validation loss. Similarly, in Figure 9, the LLM explores central and boundary points of the search space, employing a line-search-like strategy to navigate the search space.
>
> With larger compute budgets, the performance of LLMs depends more on the efficiency of the search algorithm. Table 5 demonstrates that LLMs remain competitive during extended searches (budgets of 60 and 100) across HPOBench tasks. This suggests that LLMs exhibit robustness across different computational budgets.
>
> Additionally, we demonstrate that LLMs can offer proposals when there is a mix of continuous and discrete hyperparameters in the deep learning experiments (Figure 3) and when there are conditional hyperparameters (e.g., learning rate, which is affected by the choice of optimizer). While these observations provide a starting point, we agree that understanding the mechanisms in greater depth is an open challenge. We can probe whether LLMs implicitly model function landscapes for a revision, but this appears to be an open problem requiring more investigation.
>
> > Limitations of the method.
>
> Our approach appears most effective during the initial search phase, where the computational budget is limited, and initial proposals and exploration are critical. However, techniques such as Bayesian Optimization should outperform LLMs asymptotically, as it is a global optimization algorithm. We discussed other limitations in Section 5 and how we addressed reproducibility and dataset contamination challenges.
>
> > Minor: Table 1 caption
>
> The caption for Table 1 is located above the table. We placed the table mid-page to improve the overall flow of the surrounding sections, but we can adjust this placement if preferred.

---

### Review · Reviewer_TXgS · 2024-12-10

**Summary Of Contributions:**

**Novel LLM-based HPO framework.** The paper proposes an algorithmic framework that uses large language model (LLM) for hyperparameter optimization, which repetitively prompts the LLM for better hyperparameter given the historical evaluations and configurations. The paper further examined how the chain-of-thought prompt, the information correctness of prompt, and different prompting format affect the method's performance.

**Improved numerical performance over baseline methods.** The numerical results in HPOBench and CIFAR show that the proposed method outperforms the random search and Bayesian optimization in terms of the test loss.

While the idea seems a bit straightforward, it may shed a light on a new paradigm of hyperparameter optimization.

**Audience:**

Yes

**Broader Impact Concerns:**

Not applicable.

**Claims And Evidence:**

Yes

**Requested Changes:**

Based on weakness, I suggest the author to include more comprehensive evaluation on the proposed HPO method. Specifically, the following experiments are considered as minimal to consolidate the effectiveness of the method:

1. **More challenging task.** Evaluateing the method using large language models, under the task of either finetuning, continue pre-training, or pre-training.

2. **Diverse evaluation metric.** Using different metrics other than test loss to examine whether the proposed approach can consistently improve the performance over the random parameter search and other HPO methods.

Additionally, the author should provide detailed experimental setting. The current manuscript misses some vital information for assessing the method, e.g. the model scale of the vision transformer used in CIFAR experiment is not presented.

**Strengths And Weaknesses:**

**Strength:**

- The idea of incorporating LLM heuristics into the hyperparameter optimization is novel and interesting. It does not require manual specification of the parameter search space.
- The numerical results on HPOBench and CIFAR are promising. There is a significant improvement over the random search and the Bayesian optimization baseline, which verifies that LLM can strategically generate useful search direction and search spaces.

**Weakness:**

- The evaluations are based on small models and relatively simple classification tasks. To justify the effectiveness of the proposed method, it would be critical to conduct comprehensive experiments on larger scale model and more challenging tasks, e.g. pre-training/finetuning of large language models, which are the main scenarios that requires HPO.
- Based on Table 1, the method’s performance is highly affected by the LLM’s reasoning capability. When using a relatively weak model, e.g. GPT-3.5 Turbo, its performance drops dramatically and is even worse than the random search. Since the access of powerful LLM such as GPT-4 Turbo is usually restricted, the applicability of the method is limited.
- The evaluation metric is restricted to test loss. It is critical to also justify whether the method can improve the downstream metric when performing HPO, e.g. the MT-bench score of LLM finetuning.

---

> ### Author Response · Authors · 2024-12-16
>
> Thank you for your feedback.
>
> >  The evaluations are based on small models and relatively simple classification tasks. To justify the effectiveness of the proposed method, it would be critical to conduct comprehensive experiments on larger scale models and more challenging tasks, e.g., pre-training/finetuning of large language models, which are the main scenarios that require HPO.
>
> We appreciate the suggestion to evaluate the method on larger-scale models. However, we respectfully disagree with the premise that such experiments are necessary to validate the effectiveness of our approach, as the scale of models and tasks is the current standard for papers researching HPO.
>
> First, the HPOBench framework already encompasses tasks with varying landscapes and smoothness levels, as Eggensperger et al. noted, providing a diverse range of scenarios for benchmarking hyperparameter optimization methods. This diversity aligns with the practical use cases where practitioners are interested in optimizing different models and tasks.
>
> Second, while larger pre-training tasks are undoubtedly interesting, they are computationally prohibitive to evaluate for this study. The inner optimization loops required for our approach and the baselines would significantly exceed our resources. However, if the reviewer has specific setups or configurations in mind, we may be able to demonstrate that the initial configurations suggested by the LLM yield reasonable performance.
>
> >  The evaluation metric is restricted to test loss. It is also critical to justify whether the method can improve the downstream metric when performing HPO, e.g., the MT-bench score of LLM finetuning.
>
> If we understand this concern correctly, we want to clarify that HPO aims to optimize the validation metric (of the final trained model) specified in the problem definition. In our experiments, we evaluated the validation accuracy (or error rate) for HPOBench tasks and test loss for CIFAR experiments, which are standard proxies for generalization. Whether improvements in these metrics are generalized to downstream metrics depends on the design of the optimization problem, which falls outside the direct scope of HPO performance. Instead, it is a question of how you design the problem you are optimizing.
>
> > Since the access of powerful LLM such as GPT-4 Turbo is usually restricted, the applicability of the method is limited.
>
> We have incorporated experiments with the open-source Llama-3 model in our revision, where it attains similar performance in Figure 6. This helps address the issue of cost.
>
> > The author should provide more detailed experimental setting.
>
> Thanks for pointing this out; we have added it to the paper.
>
> ``The Vision Transformer (ViT) model we use has an input image size of 32x32 pixels, divided into patches of 4x4. It has a hidden dimension of 512, a depth of 6 layers, and 8 attention heads. The model has MLP dimensions of 512, and both dropout and embedding dropout rates are set to 0.1.''

---

> ### Comment · Reviewer_TXgS · 2024-12-26
> **Follow-up Concerns**
>
> Thank you for your response. After reviewing the authors' comments, I still have the following concerns:
>
> * **Necessity of larger-scale experiments:** Conducting experiments on larger-scale models is crucial for demonstrating the practical usefulness of the proposed algorithm. The benchmarks used in the paper are small-scale and focus primarily on classical machine learning algorithms (e.g., logistic regression, SVM). Optimizing these models is significantly easier compared to modern large-scale language models, which represent the primary scenarios where HPO is most needed.
>
>   * **Specific experimental suggestion:** While I understand that repeatedly training large-scale models is computationally prohibitive, it should be feasible to experiment with smaller models. For example, you could pre-train a LLaMA model with 9 million parameters (or even 2 million) on the C4 dataset and validate its convergence. This model is considerably smaller than those typically fine-tuned in practice and should be computationally efficient. You may refer to the repository in [1] for implementation guidance.
>
> * **Evaluation beyond loss and accuracy:** It would be more compelling to evaluate additional metrics beyond loss and accuracy. Since downstream performance heavily depends on hyperparameters, it should also fall within the scope of HPO. Testing downstream performance on tasks such as MMLU or HellaSwag would provide more comprehensive insights and should be relatively quick to implement.
>
> [1] GaLore: Memory-Efficient LLM Training by Gradient Low-Rank Projection

---

> ### Author Response · Authors · 2025-01-15
>
> Thank you for your follow-up comments. We appreciate your continued engagement with our work and the specific suggestions provided.
>
> > Necessity of larger-scale experiments
>
> Our current work encompasses neural net architectures including MLPs, CNNs, and Transformers. However, we acknowledge your interest in larger language models and will investigate conducting an experiment with a smaller-scale LLaMA model (2-9M parameters) over the next few days, working within our computational constraints.
>
> > Evaluation beyond loss and accuracy
>
> This primarily relates to the design of the hyperparameter optimization setup rather than the optimization algorithm itself. The relationship between various metrics (validation loss and performance on benchmarks like MMLU) is independent of the choice of hyperparameter optimization algorithm. When multiple metrics are of interest, it becomes a multi-objective optimization problem requiring explicit specification of trade-offs between metrics.
>
> Nevertheless, we can probe how our method's improvements in validation metrics correlate with specific downstream tasks if this is of interest.

---

### Decision · Action_Editor_s2zF · 2025-01-24

**Recommendation:** Reject

**Comment:**

This manuscript considers the use of large language models for sequential hyperparameter optimization. The authors propose a framework for achieving this goal, which they evaluate in an empirical study on benchmark hyperparameter optimization problems. The proposed method seems to outperform alternatives such as Bayesian optimization on some tasks.

The reviewers of this work agree that the topic of this manuscript would be of considerable interest to a (sizable) segment of the TMLR audience. However, the reviewers also expressed considerable concerns regarding the design of the empirical study:

- a concern that the set of baselines considered was not complete
- a concern that the set of datasets / problems considered was outdated / not sufficient
- a concern that the datasets / problems considered were broadly too "small"

Combined, these concerns raise concerns regarding whether the claims made in the manuscript are supported by convincing evidence. As a result, it is not possible to recommend acceptance to TMLR at this time.

The reviewers did provide some comments regarding how the experimental study might be improved. I encourage the authors to resubmit this manuscript for reconsideration if the concerns above can be satisfactorily addressed.

**Audience:**

Yes, the focus of this manuscript on using large language models for hyperparameter optimization, two topics that are of considerable interest to the TMLR audience.

**Claims And Evidence:**

The reviewers did not come to consensus regarding this key question. Namely, the reviewers expressed concerns regarding the scope and design of the experimental study in this work, and questioned whether the results provided clear and convincing evidence of the paper's claims.

**Resubmission Of Major Revision:**

The authors may consider submitting a major revision at a later time.